# Automated Mammogram Analysis with a Deep Learning Pipeline

**Azam Hamidinekoo**[1]*      AZH2@ABER.AC.UK
**Erika Denton**[2]      ERIKA.DENTON@NNUH.NHS.UK
**Reyer Zwiggelaar**[1]      RRZ@ABER.AC.UK

[1] *Department of Computer Science, Aberystwyth University, Aberystwyth, United Kingdom*

[2] *Department of Radiology, Norfolk & Norwich University Hospital, Norwich, United Kingdom*

## Abstract

Current deep learning based detection models tackle detection and segmentation tasks by casting them to pixel or patch-wise classification. To automate the initial mass lesion detection and segmentation on the whole mammographic images and avoid the computational redundancy of patch-based and sliding window approaches, the conditional generative adversarial network (cGAN) was used in this study. Subsequently, feeding the detected regions to the trained densely connected network (DenseNet), the binary classification of benign versus malignant was predicted. We used a combination of publicly available mammographic data repositories to train the pipeline, while evaluating the model's robustness toward our clinically collected repository, which was unseen to the pipeline.

**Keywords:** Mammography, Generative Adversarial Network, DenseNet, Detection, Segmentation, Classification

## 1. Introduction

Breast mass lesions are mostly dense and appear in grey to white pixel intensity values on mammograms, with various size, distribution, shape and density. Computer-Aided Detection (CADe) and Diagnosis (CADi) systems based on deep learning (DL) methods have been improved with regards to their performances but still cannot identify all cancerous cases (Hamidinekoo et al., 2018a). DL can be applied directly on the whole image or in a patch-based approach. Majority of the approaches proposed for mass detection (Hamidinekoo et al., 2018a) use a patch-based method for training and a sliding window approach for testing. These methods basically involve patch classification followed by localisation, which are time-consuming with substantial redundancy for overlapping patches. To avoid the computational redundancy of these approaches, we proposed using the cGAN (Isola et al., 2017) to increase the efficiency by training on whole mammographic images.

The contribution of this work was to develop an automatic detection and segmentation model (CADe model) using images from multi-centres and detect mass abnormalities on our clinically collected unseen mammograms. To complete our pipeline, the detected regions

---

* Corresponding Author

were fed into the DenseNet (Huang et al., 2017) (CADi model) to predict the benign or malignant nature of the detected regions.

## 2. Methodology

### 2.1. Datasets

Four multi-centred, publicly available mammographic databases were used to build the CADe and CADi models. The combination of them is referred to as Set-1. After building the CADe and CADi models, their performances were evaluated using an unseen dataset, called Set-2. This dataset was clinically collected from the Norfolk & Norwich University hospital, UK. The information about these datasets are summarised in Table 1. In the pre-processing step, all images were segmented into background and tissue and the intensity values of the segmented regions were normalised (Hamidinekoo et al., 2018b).

For building the CADe, an additional dataset was created from the raw images in Set-1, where Gaussian noise with $\sigma = 2$ was added to increase the number of samples.

For building the CADi, Regions of Interest (RoIs) were extracted with size equal to double the square bounding box of the lesion to include the neighbourhood information. These extractions were scaled to 256×256 followed by random crops of 224×224, which were used for training the CADi (Hamidinekoo et al., 2017).

### 2.2. Model Architecture

**CADe development** Mass detection and segmentation on mammograms can be defined as translating each image into the corresponding semantic label map representing mass lesion in the breast tissue. Therefore, we propose using the conditional GAN (cGAN) (Isola et al., 2017) in order to apply a specific condition on the input images to train a conditional generative model for the CADe. In the architecture of the utilised cGAN, a U-Net-based structure and a convolutional PatchGAN classifier were used as the generator and the discriminator, respectively. Both the generator and the discriminator used modules of the form convolution-BatchNorm-ReLu. The generator was trained to generate mask images from the mammographic images, which were expected to be similar to the mask images of the real observed images from Set-1 samples. The cGAN-based CADe model was trained on the prepared training set via the stochastic gradient descent (SGD) solver with learning rate=0.00004 and batch-size=16 (using parameter tuning) for epochs=50. When several regions were detected, connected component analysis was used in the post-processing section and the three largest detected regions were extracted as RoIs.

Table 1: Utilised databases containing mass lesions.

| | Set-1: used for training the CADe and CADi models | | | | Set-2: used for testing |
|---|---|---|---|---|---|
| | BCDR-F03 (Lopez et al., 2012) | BCDR-D01 (Lopez et al., 2012) | DDSM (Heath et al., 2001) | Inbreast (Moreira et al., 2012) | Private-Dataset |
| Number of cases | 341 | 51 | 975 | 102 | 103 |
| Number of images | 664 | 105 | 1930 | 102 | 210 |
| Benign images | 369 | 69 | 1023 | 34 | 46 |
| Malignant images | 295 | 36 | 907 | 68 | 164 |
| Resolution (bits/pixel) | 8 | 14 | 12, 16 | 14 | 12, 14 |
| Image mode | digitised | digital | digitised | digital | digital |
| View | MLO, CC | MLO, CC | MLO, CC | MLO, CC | MLO, CC |
| Age distribution | 58.4±15.3 | 57.7±13.5 | 58.9±11.5 | - | 60.9±17.7 |

**CADi development** Based on a comparative study (Hamidinekoo et al., 2018b), among the well-known deep CNNs, the DenseNet was found as an appropriate model for mass classification due to its key characteristic to bypass signals from the preceding layers to the subsequent layers. In our implementations, the DenseNet's growth rate was set to 4 to construct 4 dense-blocks and 3 transition layers in the architecture. In this model, the final Softmax classifier made a binary decision based on the created features. The rest of the model's parameters (kernel, stride and padding sizes) were kept as default (Huang et al., 2017). The objective of training for the CADi was to minimise the difference error between the network prediction and the expected output (benign vs. malignant). DenseNet was trained via the SGD solver with Gamma=0.1, momentum=0.9 and weight-decay=$10^{-5}$ along with batch-size=8 (based on our hardware specifications), a dynamic learning rate with initial value=0.001 for 30 epochs. We used transfer learning with the ImageNet dataset, whilst the network was fine-tuned using Set-1. The trained CADi was able to classify images in the validation set with the accuracy of 76% and 90% and the AUC of 0.78 and 0.87 on the combination of digitised+digital images and only digital images, respectively.

## 3. Results & Discussions

To evaluate CADe performance, each mammographic image (from Set-2) was fed to the model and the probability map corresponding to the probable detected lesion was computed and compared with their annotated images. Using the CADe, mass abnormalities on mammographic images were detected with 34% accuracy, 34% precision, 32% recall and an F.Score of 0.33. The detected RoIs were segmented with Dice Similarity Coefficient=0.33±0.30 and Hausdorff Distance=8.33±2.49. Considering these results, the detection and segmentation values did not look good. This was caused due to several predictable reasons: (1) Mass segmentation is not explicitly undertaken in regular breast screening. So, the available annotations for the Set-1 data repositories and Set-2 samples were prone to subjectivity, which was not possible to consider due to the lack of such information; (2) All the lesion boundaries that were provided by the Set-1, were roughly made. But the annotated contours that were provided for the Set-2 samples were delicate and very fine in texture and structure. The model was learned to find a rough boundary not a very fine contour, which led to low DCS and large HD values; (3) All the training images used in the Set-1, had single mass lesion on each individual image but there were a significant number of testing samples (from the Set-2) with multiple lesions annotated on them; (4) There were several new appearances in the testing samples that were not seen in the training set, like micro-calcification deposits or prosthesis as shown in Figure 1; (5) The Set-2 images were originally in DICOM format and using a thresholding approach were converted to .png format. In our experiments we discovered that the value of the threshold applied to this conversion was very important, which could affect the detection performance; (6) Comparing the lesion size distributions of the annotated vs predicted lesions illustrated that the model was able to detect smaller normalised lesion sizes in the range of (0,.1]. As shown in the examples provided in Figure 1, some lesions were partially detected but were not considered as an accepted performance in the evaluation section because of the low dice score (DCS < 0.5). According to these reasons, many of the segmented bounding boxes (in the testing set) were not accurately aligned with the annotated masses, but the preliminary

Table 2: Comparing image-based classification performance using the model detected regions vs. annotated lesions by the radiologist for 210 mammograms in Set-2.

| Patches From | Actual Label | Predicted Label | | Per-class Accuracy | Not Detected | Prediction Accuracy |
| | | Benign | Malignant | | | |
|---|---|---|---|---|---|---|
| $CADe_{(\sigma=2)}$ | Benign | 15 | 10 | 60.00% | 21 | 41.90% |
| | Malignant | 41 | 73 | 64.03% | 50 | |
| Annotated lesions | Benign | 36 | 10 | 78.26% | - | 73.33% |
| | Malignant | 46 | 118 | 71.95% | - | |

Table 3: Comparing subject-based classification performance using the model detected regions vs. annotated lesions by the radiologist for 103 subjects with biopsy proven diagnosis.

| Patches From | Actual Label | Predicted Label | | Per-class Accuracy | Wrong Detection & Diagnosis | Diagnosis Accuracy |
| | | Benign | Malignant | | | |
|---|---|---|---|---|---|---|
| $CADe_{(\sigma=2)}$ | Benign | 11 | 6 | 64.70% | 5 | 63.10% |
| | Malignant | 18 | 54 | 75.00% | 9 | |
| Annotated lesions | Benign | 9 | 11 | 45.00% | - | 78.64% |
| | Malignant | 11 | 72 | 86.74% | - | |

results on 103 subjects showed the promise of this cGAN model for initial lesion localisation to be combined in the traditional image processing techniques (i.e. initial seed point for the region growing approach).

Table 2 compares confusion matrices of image based classification results on double the square bounding box of the detected lesions (through $CADe_{(\sigma=2)}$) with the actual pre-detected mass lesions by our radiologist for Set-2. The quantitative results illustrated that the trained CADi could classify pre-detected mass patches with the accuracy of 73.3%, which demonstrated (1) its representational capacity to learn different abnormality features from various digital and digitised training samples; (2) its robustness and generalisability for an unseen data repository for the task of classification. However, comparing the classification performance for the pre-detected lesions vs $CADe_{(\sigma=2)}$ detected regions, comparative accuracy values were not obtained (73.33% on 210 images vs 41.90% on 139 images, respectively). This suggested that the model was better to be used for the classification of the lesions that were initially localised and segmented by the radiologist. Considering that during the reading of screening mammograms, radiologists use multiple views, we combined the results of different views for each patient. Giving the priority to the malignant predictions, Table 3 states the diagnosis output. These results justified the use of radiologists' annotated regions for gaining better output from this pipeline.

## 4. Conclusion

Our contribution in this paper was to implement an automated pipeline based on cGAN and DenseNet models along with the required specifications to initially analyse whole mammographic images. This model avoided the computational expense of currently used patch-based or sliding window approaches, commonly used for large size images (i.e. mammograms). This exploratory research work could be further extended to using DICOM data instead of converting them into different formats in order to keep the wealth of original captured data and decrease the model sensitivity to intensity variations.

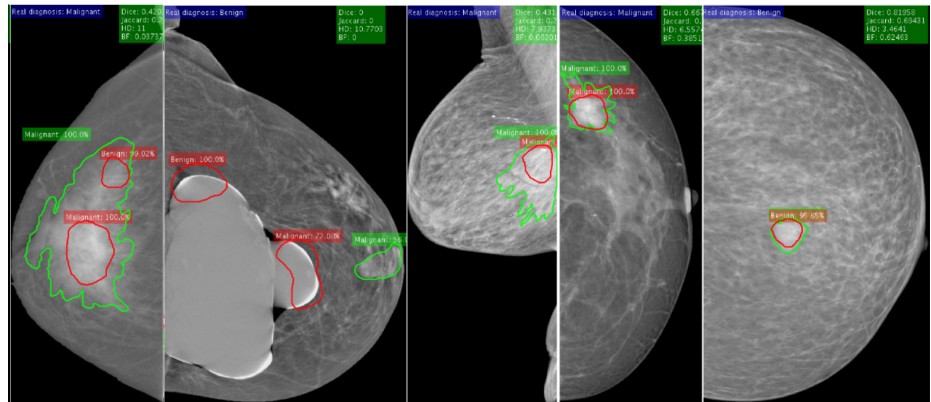

Figure 1: Examples of various detection, segmentation and classification performances on Set-2. From left to right: 3 failed detections due to low DSC and 2 accepted detections and classifications from class malignant and benign.

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
