# OpenReview forum: "Automated Mammogram Analysis with a Deep Learning Pipeline"
_MIDL.io/2019/Conference/Abstract — MIDL Abstract 2019_

### Official Review · AnonReviewer1 · 2019-04-24
**Topic is important but the novelty is limited**

**Rating:** 3
**Confidence:** 3

**Review:**

The application topic mammogram analysis is very hot in medical imaging.
The paper is well organized and with plenty of evaluation.
The author used different data resources to train the model.
However, the novelty on the method is limited.

---

### Official Review · AnonReviewer2 · 2019-04-30
**Detection and classification of lesions in the mammograms using deep learning networks**

**Rating:** 2
**Confidence:** 2

**Review:**

The method is not very clearly written and level of provided details in quite unbalanced.

For example, the authors state that the advantage is the analysis of whole images. However, they also state that regions of interest were extracted with the size equal to the double of the square of the bounding box around the lesion. It is not clear how these bounding boxes were created and in addition, I could be wrong, but this seems conflicting with the statement on the analysis of the whole mammograms. I understand that these ROIs were used only for the diagnosis, but somehow the advantage of the method seems overstated.

None of the methodological choices are motivated and some steps are not well described (e.g. transfer learning). Results seem good, but I can not relate them to other methods.

It is also unclear how the results of the multiple views were combined. Also, I wonder whether the method analyses them separately (was their one training on all views or separate training for specific views).

What is meant by Annotated lesions in Table 2 is not clear. It is also impossible to understand what is red and what green annotation in Fig 1. It would be good that these are obvious from the captions.

In addition, the abstract is too long – Table 2 and Fig 1 are on page 4.

---

### Decision · Program_Chairs · 2019-05-06
**Acceptance Decision**

Accept